# Notch and Wnt Signaling Modulation to Enhance DPSC Stemness and Therapeutic Potential

**DOI:** 10.3390/ijms24087389

**Published:** 2023-04-17

**Authors:** Verónica Uribe-Etxebarria, Jose Ramon Pineda, Patricia García-Gallastegi, Alice Agliano, Fernando Unda, Gaskon Ibarretxe

**Affiliations:** 1Medicine/Pathology Department, New York University, 550 1st Avenue, New York, NY 10016, USA; 2Cell Biology and Histology Department, University of the Basque Country (UPV/EHU), Barrio Sarriena s/n, 48940 Leioa, Spain; 3Achucarro Basque Center for Neuroscience Fundazioa Leioa, Sede Building, 3rd Floor, 48940 Leioa, Spain; 4Physiology Department, University of the Basque Country (UPV/EHU), Barrio Sarriena s/n, 48940 Leioa, Spain; 5Division of Radiotherapy and Imaging, Cancer Research UK Cancer Imaging Centre, The Institute of Cancer Research and The Royal Marsden NHS Foundation Trust, London SW7 3RP, UK; 6Department of Materials and Department of Bioengineering, Institute of Biomedical Engineering, Imperial College London, Exhibition Road, London SW7 2AZ, UK

**Keywords:** hDPSCs: human dental pulp stem cells, DNMT: DNA cytosine-5-methyltransferase, HAT: Histone acetyltransferase, H3K4me3: Histone H3 lysine 4 trimethylation, H3K9me3: Histone H3 lysine 9 trimethylation, H3K27me3: Histone H3 lysine 27 trimethylation, NNMT: Nicotinamide N-methyltransferase, iPSCs: induced pluripotent stem cells, ESCs: Embryonic Stem Cells

## Abstract

The Dental Pulp of permanent human teeth is home to stem cells with remarkable multilineage differentiation ability: human Dental Pulp Stem Cells (DPSCs). These cells display a very notorious expression of pluripotency core factors, and the ability to give rise to mature cell lineages belonging to the three embryonic layers. For these reasons, several researchers in the field have long considered human DPSCs as pluripotent-like cells. Notably, some signaling pathways such as Notch and Wnt contribute to maintaining the stemness of these cells through a complex network involving metabolic and epigenetic regulatory mechanisms. The use of recombinant proteins and selective pharmacological modulators of Notch and Wnt pathways, together with serum-free media and appropriate scaffolds that allow the maintenance of the non-differentiated state of hDPSC cultures could be an interesting approach to optimize the potency of these stem cells, without a need for genetic modification. In this review, we describe and integrate findings that shed light on the mechanisms responsible for stemness maintenance of hDPSCs, and how these are regulated by Notch/Wnt activation, drawing some interesting parallelisms with pluripotent stem cells. We summarize previous work on the stem cell field that includes interactions between epigenetics, metabolic regulations, and pluripotency core factor expression in hDPSCs and other stem cell types.

## 1. Introduction

Dental Pulp Stem Cells (DPSCs) are a population of non-differentiated cells isolated from the dental pulp of mature human teeth. These stem cells are derived from the neural crest during embryogenesis [1,2,3] and were first described in the year 2000 as a mesenchymal-like stem cell source capable of regenerating a complete dentin-pulp complex after in vivo transplantation [4,5]. Ever since, extensive research on DPSCs has revealed the extraordinary capacity of these cells to differentiate into multiple mature cell types of interest for cell therapy, including cell lineages belonging to the three embryonic layers (ectoderm, mesoderm, and endoderm) [6,7]. DPSCs constitute a highly promising source of cells for tissue engineering and regeneration, owing to their easy accessibility, high rates of ex vivo growth and self-renewal, high migration ability, low morbidity upon transplantation, high tolerance to hypoxia/ischemia, extremely versatile differentiation potential, and high tolerance to biomaterials [8,9,10,11,12,13,14]. Furthermore, the immunosuppressive and non-tumorigenic properties of DPSCs make them very well tolerated for in vivo cell grafts [15]. In the following years after the discovery of DPSCs, it was found that other stem cell types with very similar neural crest phenotypes could also be obtained from gingival and periodontal tissues [16,17,18,19,20]. However, cells from periodontal tissues are often isolated from immature teeth, and their extraction in aseptic conditions is not as simple as for DPSCs. Thus, the remarkable advantages of DPSCs with respect to other stem cell types make them very attractive candidates for cell therapy [21,22,23]. Several companies worldwide are nowadays dedicated to the DPSC banking business [24]. Compared to other more commonly used multipotent stem cell sources such as Mesenchymal Stem Cells (MSCs) from the bone marrow, DPSCs show a far better proliferative capacity [19], a better reprogramming efficiency than induced pluripotent stem cells (iPSCs) [25,26], and a broader ability for cell differentiation including neural cell lineages, particularly on serum-free media [6,19,27,28]. Nevertheless, like other adult human stem cells, DPSCs can also lose their stemness potential upon repetitive culture passages and after undergoing several cycles of cell division [29,30]. The step of ex vivo DPSC culture and expansion is often inevitable because of the relatively small amounts of biological material that can be extracted from the dental pulp of a single tooth piece. Therefore, it is crucial to research different methodologies and strategies for improving their already quite high stemness, as well as maintaining their intrinsic differentiation potential.

During embryonic development neural-crest-derived stem cells undergo epithelial-mesenchymal transition (EMT) and migrate to pharyngeal arches, to generate craniofacial tissues including the dental pulp and dental follicle of the teeth, muscles, bones, cartilages ganglia, and nerves of the peripheral nervous system [21,31]. Some of these non-differentiated neural-crest stem cells remain in the dental pulp of adult teeth, giving rise to DPSCs. These stem cells have a high potential to generate dental tissues and connective tissues [5] but also retain the ability to differentiate into different cell types including melanocytes, odontoblasts, osteoblasts, chondrocytes, adipocytes, neural progenitors, vascular endothelial and pericyte cells, smooth muscle cells, and even endodermal lineage cells [6,7,32,33,34]. A note of caution should be introduced here, as many of these studies largely rely on the expression of lineage-specific cell markers, and not so much on the physiological characteristics of the differentiated cells. Thus, there is a possibility that the differentiation phenotypes might be incomplete, despite the marker expression profile suggesting a full terminal differentiation. The case of neuronal differentiation is paradigmatic because even though neurodifferentiated DPSCs express a very complete repertoire of mature neuronal markers, including NeuN, MAP2, TUJ1, neurofilaments, synaptic proteins, neurotransmitter transporters, receptors, and voltage-dependent ion channels; there is yet no definite electrophysiological proof of their neuronal functionality. Despite their strong neuronal and synaptic marker expression, no conclusive evidence of physiologically functional synaptic coupling of neurodifferentiated DPSCs with other neuronal cells has ever been yet reported [10,27,28].

The unusually high multilineage differentiation potential of DPSCs could be explained by their neural crest origin. In fact, DPSCs share important characteristics with neural crest stem cells and progenitors, including a largely coincident pattern of gene expression, showing the expression of neural crest factors (SNAIL/SNAI1, SLUG/SNAI2, TWIST1, HNK1, PAX3, NEUROGENIN2 and, SOX10), neural markers (TUJ1, GFAP, and NESTIN) and pluripotency core factors (OCT4, CMYC, SOX2, KLF4, LIN28, REX1, SSEA1, STELLA, and NANOG) [26,35,36]. In addition, DPSCs also express a combination of MSC and fibroblastic markers (COLLAGEN I, STRO1, VIMENTIN, and α-SMA; Figure 1). MSCs and fibroblasts are highly related types of cells, with a largely coincident molecular marker expression profile [35]. Therefore, the sorting of stem and fibroblast cell populations can be technically challenging when working with cultures of surface-adherent DPSCs grown in fetal serum-containing media. These culture conditions tend to favor the mesenchymal lineage commitment of DPSCs, at the expense of a reduction in their neural differentiation potential [28].

In particular, the expression of pluripotency core factors seems to be essential for the maintenance and self-renewal of DPSCs [35]. Indeed, DPSCs express significant amounts of at least three out of four of the original pluripotency “Yamanaka” factors [25,26,35,36] with reportedly variable expression levels for KLF4 [26,35]. These factors participate in a complex pluripotency network principally formed by a core set of embryonic stem transcription factors (OCT4/POU5F1, SOX2, NANOG, LIN28, among others) [37,38]. REX1, SSEA1, STELLA, CMYC, and KLF4 have also been reported to participate in this pluripotency network [37]. CMYC is a transcriptional target of the key pluripotency factors OCT4 and NANOG and its activity increases the rate of proliferation by helping to establish a pluripotent cell cycle in reprogrammed cells [39,40]. Additionally, KLF4 might contribute to the activation of NANOG and other pluripotency-related genes [39]. Altogether, the expression of pluripotency core factors seems to greatly facilitate the reprogramming of DPSCs to iPSCs [25,26,41,42]. DPSCs have been reported to generate cells from the three embryonic layers, including their capability of trans-differentiation to epithelial/endothelial cells [43,44]. The pluripotency of DPSCs has been the subject of a very interesting debate, where some authors reported the generation of teratomas after DPSC transplantation into host nude mice [7]. More than a decade later, these data have still not been replicated. In addition, the final requisite to demonstrate cell pluripotency (i.e., implantation to embryo blastocysts and generation of viable chimera organisms) was also reported to be partially met by DPSCs more than a decade ago, but these results have neither been yet replicated [45].

## 2. Metabolic and Epigenetic Remodeling in Stem Cells

Cell metabolism is known to play a significant role in the production of energy, cell-fate determination, and stem cell activity in different contexts [46,47]. Accordingly, the first set of activated genes during cell reprogramming corresponds to genes involved in increased proliferation and metabolic remodeling. Cell metabolism can be divided into aerobic/oxidative, completed in the mitochondria, or anaerobic/glycolytic, completed in the cytosol. Importantly, the expression of glycolysis-related genes and lactate production in Embryonic Stem Cells (ESCs) and adult multipotent stem cells is higher compared to terminally differentiated cells, which rely preferentially on mitochondrial oxidative phosphorylation (OXPHOS) to fulfill their energy needs [48]. Oxidative phosphorylation is used by somatic cells for energy production, but during cell reprogramming, the transition to pluripotency is accompanied by a shift to glycolytic metabolism, and in fact, the efficiency of cell reprogramming is significantly enhanced by raising glucose concentrations in the culture medium [49,50]. It should also be noted that many components of the core pluripotency factor network crucially regulate cellular metabolism [46,51]. For instance, the well-studied core pluripotency factors OCT4 and CMYC regulate glycolysis directly by targeting key glycolytic enzymes in ESCs, such as HK2 and PKM2, which affect lactate, acetate, and glucose metabolites. CMYC also regulates metabolic gene targets, influencing glycolysis, mitochondrial biogenesis, and glutamine and proline catabolism, among others [46,52,53]. The pluripotency core factor LIN28 has also been reported to play a role in regulating metabolism by turning off mitochondrial OXPHOS and enhancing one-carbon metabolism and histone methylation [54]. However, further research is necessary to acquire a comprehensive understanding of the effects that pluripotency core factors carry out in cellular metabolic remodeling.

In the early stages of embryonic development pluripotency core factor genes remain active, but at later stages of cell differentiation they gradually become silenced, and cell type-specific genes are turned on instead. This is the result of the expression of specific transcription factors associated with chromatin modifications and remodeling. Chromatin in ESCs displays an open conformation featured by high levels of acetylated histones, and the presence of bivalent gene promoters containing both enhancing and repressing histone methylation tags, enabling a rapid expression of lineage-specific genes during differentiation [55]. DNA methylation at CpG islands is another important epigenetic silencer mechanism [56,57]. The silencing of certain genes by DNA methylation is required for the induction of differentiation of ESCs whereas DNA hypomethylation is required for the maintenance of cell stemness [58]. Classical pluripotency core factors OCT4, CMYC, NANOG, SOX2, KLF4, and LIN28 have been shown to interact with and regulate the epigenetic landscape [59,60]. For instance, hypomethylation of the OCT4 gene promoter is known to enhance its expression, and OCT4 may in turn interact with several polycomb group proteins to regulate the histone methylation status of other genes and pluripotency factors such as NANOG [61,62,63]. A high expression of pluripotency core factors induces somatic cell dedifferentiation, making DNA more accessible and facilitating the expression of stemness-related genes [63,64].

## 3. Molecular Epigenetic and Metabolic Network in Cell Reprogramming

We are only beginning to uncover the mechanisms whereby cellular metabolism influences stem cell fate, but this includes changes at the epigenetic level, which in turn affect gene expression. According to these findings, some metabolic intermediates may contribute to the regulation of chromatin accessibility. Several studies place glycolysis as a crucial producer of metabolic intermediates necessary to produce new biomass to sustain cell growth and division and as a source of small organic methyl and acetyl groups required to perform epigenetic changes [46,52,65,66,67]. Research over the last decade has identified essential roles for small organic metabolites in the regulation of epigenetics and transcription, including S-adenosyl methionine (SAM) produced via the one-carbon cycle, Acetyl-CoA from glycolysis, as well as α-ketoglutarate (α-KG), nicotine adenine dinucleotide (NAD) and flavin adenine dinucleotide (FAD) from the Tricarboxylic Acid Cycle (or TCA cycle) [68,69,70]. Increased SAM levels lead to higher histone methylation, whereas increased ascorbate and α-ketoglutarate levels are related to histone demethylation and the maintenance of pluripotency [68,70]. N-methyl transferase enzyme (NNMT) controls SAM–SAH (S-adenosyl homocysteine) conversion, which critically regulates SAM levels, and hence the activity of DNA and histone methylases [69] (Figure 2). Interestingly, α-ketoglutarate, a metabolite derived from glutamine, is described to be a substrate for several histone demethylases, and the Tet family of enzymes that are involved in histone and DNA demethylation to maintain a relaxed chromatin state [70]. Acetyl-CoA is a versatile metabolite that can contribute to the initiation of the TCA cycle, the “de novo” biosynthesis of lipids, and the acetylation of specific amino-acid residues (predominantly lysine) on histone and non-histone proteins [71]. The mutual dependence between Acetyl-CoA and histone acetylation has been previously described in many different types of stem cells such as ESCs [72], skeletal muscle stem cells [73], and DPSCs [74,75]. Accordingly, the block of Acetyl-CoA production causes a reduction of histone acetylation and the onset of cell differentiation [52,72,76]. It has been described that the addition of Acetyl-CoA prevented histone deacetylation and delayed differentiation of ESCs in its initial stages [72]. The specific mechanisms by which metabolism can influence stem cell fate are still not fully understood, but it is known that the accumulation of some specific metabolites such as reactive oxygen species (ROS) and SAM can trigger an epigenetic remodeling, thus influencing stem cell fate [68,77]. Accordingly, some metabolic intermediates may contribute to the regulation of chromatin remodeling during cell reprogramming. This is crucial to understand the physiological events taking place during cell reprogramming, with a view to inducing a safe and controlled de-differentiation of cells to use in future cell therapy.

## 4. Notch and Wnt Enhancement of DPSC Stemness Comes along with a Metabolic and Epigenetic Remodeling

Wnt and Notch pathways are critically involved in the control of self-renewal and differentiation in many stem cell types, including DPSCs [35,78,79,80,81,82,83,84]. Therefore, a thorough understanding of the contribution of Notch/Wnt signaling pathways to DPSC maintenance is interesting for their use in cell therapy and tissue engineering.

Research reports show a complex role of Wnt/β-catenin activity in the regulation of DPSC migration, osteogenic differentiation, and mineralization [35,80,85]. GSK3β can regulate the canonical Wnt pathway through β-catenin phosphorylation and degradation in the proteasome. Wnt activation causes the inhibition of GSK3β, leading to β-catenin accumulation and translocation to the nucleus, where it activates Wnt target genes. Among the upregulated genes there is JAGGED1, which in turn activates Notch signaling [83]. β-catenin also interacts with the intracellular part of the Notch receptor [86]. Notch and Wnt pathways are also functionally interconnected in DPSCs. After activation of the Wnt signaling pathway, either by pharmacological inhibitors of GSK3β (BIO) or by canonical Wnt ligands (WNT-3A), the expression of many key pluripotency core factors (SOX2, OCT4/POU5F1, LIN28, NANOG, REX1, SSEA1) rises dramatically in DPSC cultures, leading to an increased proliferation and self-renewal capacity, enhanced maintenance of the stem phenotype, and a more efficient generation of mature cell lineages such as adipocytes and osteoblasts [35]. It is noteworthy that this upregulation of the pluripotency core factor network and enhancement of DPSC properties could be carried out by just short-term pharmacological treatments, such as 48 h stimulations with WNT-3A in vitro [35]. Thus, Notch/Wnt activation increases the expression of pluripotency core factors and enhances the stemness of DPSCs. A thorough analysis of the metabolic and epigenetic changes that took place during this stimulation gave rise to interesting findings. The first noteworthy aspect is that Wnt pathway stimulation with both BIO and WNT-3A increased cellular consumption of glucose and glutamate/glutamine by DPSCs [74]. This very clear metabolic fingerprint was accompanied by a higher mitochondrial activity (increased mitochondrial membrane potential, and higher expression of ETC subunits), a higher metabolic turnover of NADH/NAD^+^, and an increased formation of lipid droplets [74]. Interestingly, lactate levels were not affected after Wnt activation, and mitochondria were found to be hyperpolarized after Wnt stimulation, which suggested that the glycolytic end-products were being funneled to mitochondria to feed the TAC and ETC, but not necessarily to increase the synthesis of ATP. Likely, this increased metabolic influx to mitochondria would be directed to sustain the generation of citrate, which can be rapidly exported to the cytosol by cataplerosis [87]. Later, cytosolic citrate can be converted to Acetyl CoA, to promote nuclear histone acetylation (Figure 2). This view is also supported by the fact that the expression levels of cytosolic Acetyl-CoA synthesizing enzymes were also clearly increased in Wnt-stimulated DPSC cultures [74]. Finally, the remaining Acetyl-CoA can also be metabolized to form lipid droplets which could serve as an additional Acetyl-CoA storage reserve for another wave of histone acetylation [52,66].

Interestingly, a similar metabolic remodeling has also been observed in other stem cell types apart from DPSCs. For instance, it has been described that mitochondrial remodeling occurs during the induction of cell reprogramming in a phenomenon called “mitochondrial rejuvenation” [88,89,90]. These findings could explain the initial increase in the expression of mitochondrial proteins in cells undergoing this metabolic remodeling [91]. This transient “hyper-energetic” state seems to be required for cell stemness maintenance, which is accompanied by an increment in ROS production and enhancement of the glycolytic rate. Some authors have described the involvement of ROS generation in cell reprogramming [77]. Cellular Acetyl-CoA can be used by the TCA in mitochondria as a source of energy, and alternatively stored in the form of lipid droplets for its later use [52,66,89]. All these physiological changes seem to be crucial for the maintenance of high levels of histone acetylation, which are characteristic of pluripotent and multipotent stem cells. However, further studies are necessary to further clarify the role of metabolism in stemness maintenance and its relationship with pluripotency core factors.

In parallel to metabolic changes, it was demonstrated that true epigenetic remodeling also took place in DPSC cultures after Wnt pathway activation [75]. NMR analysis of global 5 mC levels demonstrated that DNA methylation decreased after Wnt treatments, giving further evidence about the plasticity of chromatin in this cell population. As for chromatin modifications, H3K4me3, H3K27me3, and H3K9me3 levels were found to be higher in BIO and WNT-3A treated DPSCs [75]. As mentioned before, increased both enhancing and repressing histone marks could be suggesting the formation of bivalent gene promoter domains containing H3K4/H3K27 [75]. The presence of bivalent promoters is highly characteristic of pluripotent stem cells, and their principal role is enabling a fast response in gene regulation of expression and repression in response to different stimuli [92]. Considering that Wnt-stimulated DPSCs showed an increase in methylation and acetylation histone marks, together with global DNA demethylation, we are inclined to think that these unequivocal epigenetic signatures are reflecting a true enhancement of the stemness of DPSC to approach a remodeling in chromatin structure (Figure 3). Taking all the above into consideration, it seems that epigenetic remodeling may play a fundamental role in future stem cell-based reparative medicine, despite more studies being necessary to expand the information on this phenomenon and its therapeutic implications.

## 5. Combination of Notch and Wnt Activation of DPSCs with Serum Free Media and Scaffolds

With all the above-mentioned evidence, it looks that DPSCs could someday become a promising source of pluripotent-like stem cells alternative to iPSCs for cellular therapy. In fact, some studies have demonstrated a superior reprogramming ability of DPSCs, compared to other stem cell types [25,26,42]. However, in order to maximize cell survival and minimize graft rejection, it would be crucial to stably grow and expand DPSC cultures in serum-free media. It is known that FBS-containing media readily induce the progressive long-term commitment of DPSCs to differentiation to osteogenic lineages after a high number of culture passages [93,94], and even young DPSC cultures always show some degree of osteoblastic pre-differentiation when grown with FBS, as assessed by the expression of pre-osteoblastic markers such as ALP, and RUNX2 [21,75]. Several articles demonstrated that even intracellular serum traces may affect their cellular immunosuppressive properties, increasing the chances of graft rejection or an immune response leading to inflammation [93,95]. Even in the case of iPSCs, the presence of serum has been reported to negatively affect their reprogramming potential [96]. For all these reasons, the use of serum for clinical therapies should be avoided. However, finding a neutral culture medium that enables cell culture expansion and survival, while not inducing any cell differentiation, has become more challenging than expected. Recently, we described that serum-free media allowed DPSC survival while enhancing the expression of endothelial cell markers and characteristics of vascular cells, promoting neovasculogenesis both in vitro and in vivo [34,43]. Other alternatives, such as the use of StemPro^®^ (Gibco, Karlsruhe, Germany) media have been shown to increase DPSCs stemness and cellular characteristics of neural crest, but at the expense of higher cell quiescence and lower proliferation rates [36]. However, the compounds and factors involved in these commercial cell culture media are often undisclosed, which complicates further research.

Another promising way to modulate cellular signaling in stem cells in general and DPSCs, in particular, relies on structural contacts using scaffolds. A large choice of two-dimensional (2D) or three-dimensional (3D) bioresorbable scaffolds is available to facilitate correct cell and tissue integration in regenerative cell therapies [97,98]. Moreover, 2D scaffolds that modulate Notch/Wnt activity are useful to study general culture characteristics, including optimal dose, activation kinetics, and cell behavior, without the complexity of the 3D microenvironment. One example is 2D nanostructured poly(l-lactide-*co*-caprolactone) (P(LLA-CL) scaffolds whose composition is based on bioresorbable polymers. This nanopatterned polymer has proven its effectiveness for the culture of neural stem cells (NSCs) [99]. Interestingly, it also allows cell alignment and directionality of migration, thus improving and accelerating neuronal differentiation when it is synthesized with graphene oxide [99], and can also be combined with DPSCs [100]. These and other scaffolds can be loaded with cells and integrate compounds that regulate Notch/Wnt pathways [101]. Thus, the combination of immobilized Notch/Wnt ligands in 2D/3D scaffolds together with stem cells can be a promising avenue for the development of enhanced tissue engineering devices. For example, the immobilization of JAGGED1 in hydrogels has been reported to create a biomaterial with novel osteo-inductive properties for bone tissue engineering [102]. Other 3D biomaterials such as decellularized extracellular matrices of different tissues also offer an alternative to studying a cellular response that could better mimic the natural 3D environments through cell-to-cell and cell–matrix interactions, to improve the future success of tissue-engineered grafts [103,104]. Some of these materials have already been successfully combined with DPSCs [105]. In this regard, the incorporation of DPSCs together with Notch/Wnt ligands could be also envisaged using other injectable biological hydrogels such as those of hyaluronic acid, with excellent biocompatibility and tissue integration properties [106,107].

## 6. Conclusions

The very remarkable multilineage differentiation potential of DPSCs has led some researchers to consider them as pluripotent-like cells [7,45,108,109,110,111] Although it is not strictly speaking possible to state that these cells have pluripotency capabilities, there is no doubt that DPSCs share a lot of characteristics with other pluripotent stem cell types such as ESCs and iPSCs. Many of the critical factors participating in the pluripotency-sustaining network are prominently expressed in DPSCs. In addition, the metabolic and epigenetic regulatory mechanisms of pluripotency seem to be highly conserved in DPSCs, with many parallelisms between these and other pluripotent stem cell types. Furthermore, controlled growth of DPSCs under conditions of stimulation of selective signaling pathways such as Wnt or Notch in serum-free media can reinforce the pluripotency-related features of these cells without the need for genetic manipulation, opening new possibilities for their use in cell therapy.

## Figures and Tables

**Figure 1 ijms-24-07389-f001:**
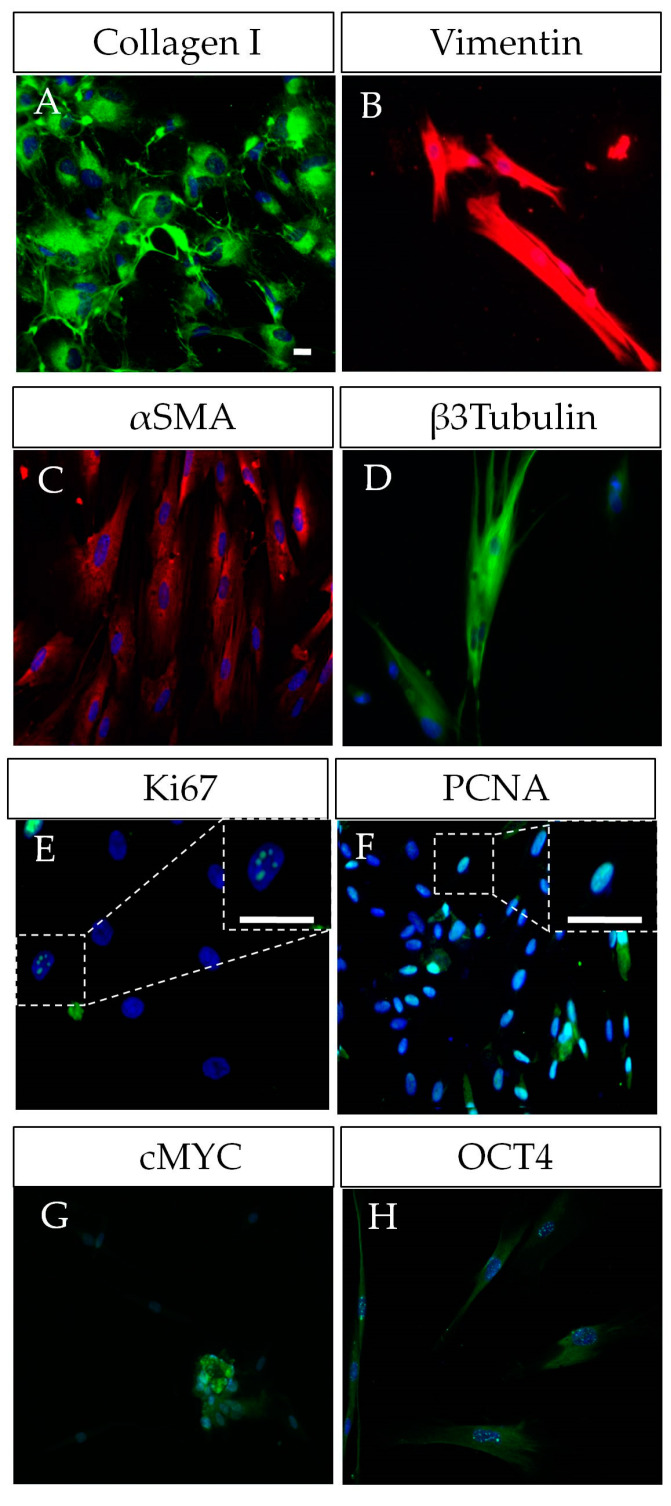
Characterization of human DPSCs in vitro. (**A**–**D**) DPSCs grown in plastic-adherent conditions show a co-expression of mesenchymal markers (COLLAGEN I, VIMENTIN, α-SMA) and neural markers (β-III-TUBULIN/TUJ1, NESTIN), this duality reflecting their neural crest origin. (**E**–**H**) DPSCs as proliferative cells also show expression of Ki67 and PCNA, and even of some pluripotency core factors such as OCT4 and CMYC, SOX2. Scale bars = 20 µm.

**Figure 2 ijms-24-07389-f002:**
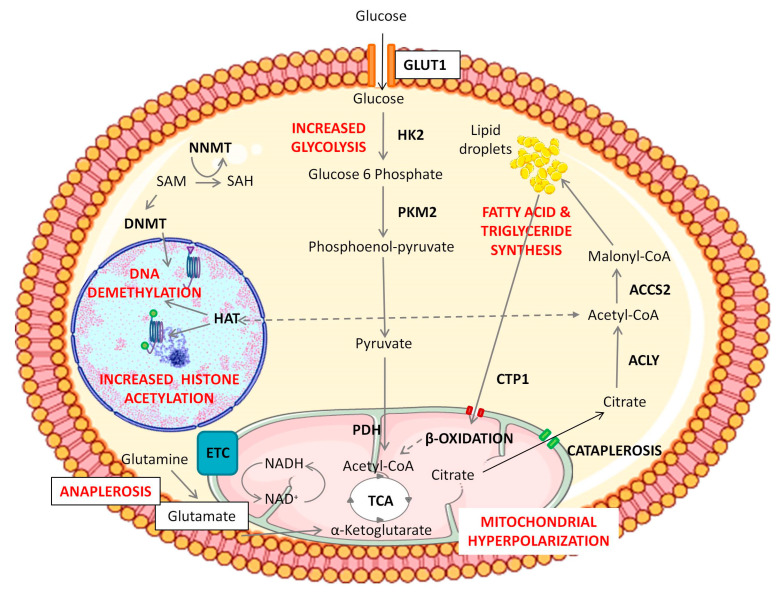
Representation of epigenetic and metabolic changes during Notch/Wnt activation in DPSCs. Notch/Wnt signaling activation triggers an increase in glucose consumption by overexpression of glycolytic enzymes HK2 and/or PKM2. Pyruvate dehydrogenase complex subunits are also upregulated in BIO/WNT-3A treated DPSCs and this leads to the generation of Acetyl-CoA to fuel the mitochondrial TCA cycle. These “hyper-energized” DPSCs show a net accumulation of lipids and a mitochondrial hyperpolarization, implying a higher activity of the TCA cycle. Overexpression of cytosolic ACLY and ACSS2 enzymes also suggests transport of citrate from mitochondria to the cytosol (cataplerosis) eventually leading to a cytosolic accumulation of Acetyl-CoA, which can be then used to sustain higher levels of histone acetylation, also supported by a HAT overexpression. Meanwhile, mitochondria consume amino acids such as glutamine and glutamate to replenish the exported TCA metabolites in a coordinated cycle of cataplerosis and anaplerosis. Cytosolic fatty acids also appear to participate in the maintenance of this cycle, as suggested by the overexpression of mitochondrial fatty acid transporters (CPT1) and β-oxidation enzymes under Notch/Wnt stimulation. Thus, de novo synthesized lipids could play a role as a storage reserve of Acetyl-CoA, by feeding the mitochondrial TCA cycle and inducing citrate cataplerosis. DPSCs thus show a boost in glycolysis together with stimulation of mitochondrial activity but without cytosolic lactate accumulation. The goal of that metabolic switch seems to provide for high levels of histone acetylation to maintain DPSCs in a non-differentiated state. Changes are also observed at DNA and histone methylation levels in Notch/Wnt stimulated DPSCs. The Figure was partly generated using Servier Medical Art, provided by Servier, licensed under a Creative Commons Attribution 3.0 (CCA3) unported license.

**Figure 3 ijms-24-07389-f003:**
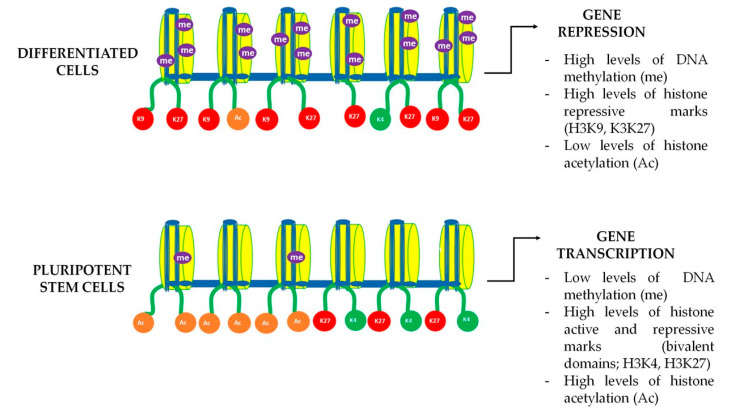
DNA bivalent domains dynamics in differentiated and pluripotent cells. Gene expression in stem cells is a dynamic process that is subject to chromatin remodeling. Chromatin dynamics and accessibility depend on the activation and repression marks known as H3K4 and H3K9 and H3K27 respectively. Pluripotent stem cells are known to have both active (green) and repressive (red) methylation marks in the same histone domain, enabling to trigger a fast gene transcription. During cell differentiation those bivalent marks are progressively lost leading to a more stable chromatin structure, only allowing the expression of a limited set of genes that are specifically necessary for mature cellular functions.

## Data Availability

Not applicable.

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
