# Peer review of "Notch and Wnt Signaling Modulation to Enhance DPSC Stemness and Therapeutic Potential"

_ijms, 2023, doi:10.3390/ijms24087389_

Round 1

Reviewer 1 Report

In the review, the authors integrated important aspects related to the maintenance of the stemness of dental pulp stem cells (DPSCs), focusing on the Notch/Wnt activation. For this, the authors summarized relevant aspects involved in the genetic, epigenetic, and metabolic regulation of pluripotency. This review is very well written in provides details, especially about the pluripotency metabolic regulation.  I strongly recommend this review for publication. 

Author Response

Manuscript: ijms-2338727

Dear Editor and Reviewers,

Thank you very much for your feedback, and your overall positive assessment of our review manuscript. We are glad to submit our revised manuscript with changes highlighted over a yellow background. Some references have been added and others removed to give a deeper insight and a better clarity to the raised arguments. We hope you will appreciate our efforts to improve the review. Thank you very much for your cooperation and constructive comments.

Reviewer 1.

In the review, the authors integrated important aspects related to the maintenance of the stemness of dental pulp stem cells (DPSCs), focusing on the Notch/Wnt activation. For this, the authors summarized relevant aspects involved in the genetic, epigenetic, and metabolic regulation of pluripotency. This review is very well written in provides details, especially about the pluripotency metabolic regulation.  I strongly recommend this review for publication.

Thank you very much indeed for your very positive comments and evaluation. We have added some small but important updates of the text highlighted over a yellow background. We genuinely hope the manuscript will be of interest to the stem cell research community.

Reviewer 2 Report

In the review: “Notch and Wnt signaling modulation to enhance DPSC stemness and therapeutic potential”, the authors discussed about  the mechanisms responsible for stemness maintenance of DPSCs, and how these are regulated by Notch/Wnt activation, drawing some interesting parallelisms with pluripotent stem cells.

Overall, this manuscript results very interesting, the authors clearly explain the rational of the study and discussed the topic point by point.

However, we would like to invite the authors  to clarify some minor points: 

11.     Please check the check punctuation and spaces;

22.  Among the introduction, the authors described in general the features of dental pulp stem cells, please better describe the advantages of DPSCs with respect to other stem cell types;

33. The  methodologies and strategies to improve DPSCs stemness are related also to other stem cell types? If not, why?;

44. Page 8, lines 310-311: in which way the  serum containing media induce the progressive commitment of DPSCs to differentiation to osteogenic lineages?

55.  The authors wrote about the expression of specific fibroblast and staminal markers, is there any indication in the literature on how they should be expressed? in what proportion to say that the cells are differentiated or not? Please deepen this concept, perhaps with appropriate reference;

66. Page 8, lines 326-347: the authors wrote about 2 and 3D scaffold for DPSCs in vitro culture, maybe also the role of hydrogels should be introduced, in particular hyaluronic acid based hydrogels;

Author Response

Manuscript: ijms-2338727

Dear Editor and Reviewers,

Thank you very much for your feedback, and your overall positive assessment of our review manuscript. We are glad to submit our revised manuscript with changes highlighted over a yellow background. Some references have been added and others removed to give a deeper insight and a better clarity to the raised arguments. We hope you will appreciate our efforts to improve the review. Thank you very much for your cooperation and constructive comments.

Reviewer 2.

In the review: “Notch and Wnt signaling modulation to enhance DPSC stemness and therapeutic potential”, the authors discussed about  the mechanisms responsible for stemness maintenance of DPSCs, and how these are regulated by Notch/Wnt activation, drawing some interesting parallelisms with pluripotent stem cells. Overall, this manuscript results very interesting, the authors clearly explain the rational of the study and discussed the topic point by point. However, we would like to invite the authors  to clarify some minor points: 

Thank you very much for your comments. We proceed to give our point-by-point answers below:

  1. Please check the check punctuation and spaces

We have searched throughout the manuscript and found a few of those mistakes, which are corrected in the new version.

  1. Among the introduction, the authors described in general the features of dental pulp stem cells, please better describe the advantages of DPSCs with respect to other stem cell types

We have added new information and bibliography in the Introduction section, where we specifically compare the characteristics of DPSCs with the more widely known Bone Marrow Mesenchymal Stem Cells. DPSCs show higher proliferation rates and better reprogramming efficiency, together with a better neural differentiation capacity. These differences can be explained by the neural crest origin of DPSCs.

New text added to the manuscript:

Compared to other more commonly used multipotent stem cell sources like Mesenchymal Stem Cells (MSCs) from the bone marrow, DPSCs show a far better proliferative capacity [19], a better reprogramming efficiency to induced pluripotent stem cells (iPSCs) [25,26], and a broader ability for cell differentiation including neural cell lineages, particularly on serum-free media [6,19,27,28].

  1. The methodologies and strategies to improve DPSCs stemness are related also to other stem cell types? If not, why?

The pharmacological regulation of Notch and Wnt pathways with compounds like BIO, DAPT and WNT-3A has been shown to critically affect cell differentiation and reprogramming in different stem cell types (Clevers et al. 2014, doi: 10.1126/science.1248012 ; Ichida et al. 2014, doi : 10.1038/nchembio.1552; Kitajima et al. 2016, doi: 10.1016/j.exphem.2015.09.007). In particular, Wnt activity is known to be essential to maintain pluripotency (ten Berge et al. 2011; doi: 10.1038/ncb2314 ; Jang et al. 2019, doi : 10.1371/journal.pbio.3000453). Concerning the particular strategy that we employed to enhance the stemness of DPSC cultures, consisting of short-applications of BIO/WNT-3A for 48 hours, this had its origin on a paper we published in 2017 (Uribe-Etxebarria et al. 2017 doi: 10.22203/eCM.v034a16.) In that study, we found that this short-term Wnt activation had a profound impact on the levels of expression of pluripotency core factors by DPSCs. Thus, we stuck to that particular model of pharmacological stimulation for metabolic and epigenetic assessments in subsequent studies.

  1. Page 8, lines 310-311: in which way the  serum containing media induce the progressive commitment of DPSCs to differentiation to osteogenic lineages?

The osteo/odontogenic lineage of cell differentiation is the preferred one by DPSCs, when these cells are grown under standard culture conditions containing FBS (Yu et al. 2010, doi: 10.1186/1471-2121-11-32; Pisciotta et al. 2012; DOI: 10.1371/journal.pone.0050542). This default pathway of cell differentiation can be regarded as a natural fate of these cells, as one of the principal  biological functions of DPSCs is to differentiate to odontoblasts and generate a protective layer of reparative dentin against harmful (e.g. carious) stimuli to the dental pulp. The osteogenic differentiation of DPSCs is enhanced by mesenchymal surface-adherent culture conditions, as opposed to growth as spheroids, which preserve better the primogenic neural crest characteristics of these cells (Luzuriaga et al. 2021. doi: 10.3390/ijms22073546). We have added new information to that paragraph, to clarify that point.

New text added to the manuscript:

It is known that FBS containing media readily induce the progressive long-term commitment of DPSCs to differentiation to osteogenic lineages after a high number of culture passages [93,94], and even young DPSC cultures always show some degree of osteoblastic pre-differentiation when grown with FBS, as assessed by the expression of pre-osteoblastic markers like ALP, and RUNX2 [21,75].

  1. The authors wrote about the expression of specific fibroblast and staminal(stem?) markers, is there any indication in the literature on how they should be expressed? in what proportion to say that the cells are differentiated or not? Please deepen this concept, perhaps with appropriate reference

An interesting point that we also address in the new manuscript version. Mesenchymal stem cells and fibroblasts are surprisingly similar cell types, and it is often difficult to sort them exclusively on the basis of expression of a particular marker (Chang et al. 2014; doi: 10.1159/000363035). This is especially challenging in the case of surface markers for flow cytometry or FACS isolation, which are largely coincident in MSCs and fibroblasts. Of course, there is the expression of pluripotency core factors and some other transcription factors like STRO-1, which are only expressed by stem cells, and not by fibroblasts. But even this differential expression of stem cell markers has begun to be challenged by some studies, and these stem-related markers are often intracellular, making the separation of stem cell and fibroblasts populations difficult in practice. We include new information together with the aforementioned reference in the new ms version.

New text added to the manuscript:

MSCs and fibroblasts are highly related types of cells, with a largely coincident molecular marker expression profile [35]. Therefore, the sorting of stem and fibroblast cell populations can be technically challenging when working with cultures of surface-adherent DPSCs grown in fetal serum containing media. These culture conditions tend to favor the mesenchymal lineage commitment of DPSCs, at the expense of a reduction in their neural differentiation potential [28].

  1. Page 8, lines 326-347: the authors wrote about 2 and 3D scaffold for DPSCs in vitroculture, maybe also the role of hydrogels should be introduced, in particular hyaluronic acid based hydrogels.

An interesting insight on a particularly promising biomaterial that can be injected and also be easily functionalized with Notch/Wnt pharmacological regulators and other compounds of interest. We include mention to HA hydrogels in that section, together with a couple of new references.

New text added to the manuscript:

Some of these materials have already been successfully combined with DPSCs [105]. In this regard, the incorporation of DPSCs together with Notch/Wnt ligands could be also envisaged using other injectable biological hydrogels like those of hyaluronic acid, with excellent biocompatibility and tissue integration properties [106,107].
